# Epigenetic Rewiring of Metastatic Cancer to the Brain: Focus on Lung and Colon Cancers

**DOI:** 10.3390/cancers15072145

**Published:** 2023-04-04

**Authors:** Annamaria Morotti, Francesco Gentile, Gianluca Lopez, Giulia Passignani, Luca Valenti, Marco Locatelli, Manuela Caroli, Claudia Fanizzi, Stefano Ferrero, Valentina Vaira

**Affiliations:** 1Division of Pathology, Fondazione IRCCS Ca’ Granda Ospedale Maggiore Policlinico, 20122 Milan, Italy; annamaria.morotti@unimi.it (A.M.); francesco.gentile0992@gmail.com (F.G.); gianluca.lopez@unimi.it (G.L.); stefano.ferrero@unimi.it (S.F.); 2Department of Pathophysiology and Transplantation, University of Milan, 20122 Milan, Italy; luca.valenti@unimi.it (L.V.); marco.locatelli@unimi.it (M.L.); 3Precision Medicine Lab, Biological Resource Center, Department of Transfusion Medicine, Fondazione IRCCS Ca’ Granda Ospedale Maggiore Policlinico, 20122 Milan, Italy; giulia.passignani@policlinico.mi.it; 4Division of Neurosurgery, Fondazione IRCCS Ca’ Granda Ospedale Maggiore Policlinico, 20122 Milan, Italy; manuela.caroli@policlinico.mi.it (M.C.); claudia.fanizzi@policlinico.mi.it (C.F.); 5Department of Biomedical, Surgical, and Dental Sciences, University of Milan, 20122 Milan, Italy

**Keywords:** brain metastases, methylation, epigenetics, lung cancer, colorectal cancer

## Abstract

**Simple Summary:**

The epigenetic state within cells represents a layer for controlling homeostasis and cell differentiation. In cancer, this fine-tuned program is disrupted, resulting in cancer progression and, eventually, dissemination to distant organs. Here, we examine the epigenetic states of brain metastasis from colorectal and lung cancers and we compare those signatures with the ones detected in primary tumors. Our aim is to decipher which are the tumor type-specific regions whose deregulation is involved in metastatic dissemination to the brain and which signaling potentially confers an advantage to tumor cells to colonize the brain milieu. This information will shed light on the epi-mechanisms underpinning CRC and LuC cell dissemination to the brain and provide preliminary clues about the potential clinical value of epigenetics for brain metastasis diagnosis and therapeutic targeting.

**Abstract:**

Distant metastasis occurs when cancer cells adapt to a tissue microenvironment that is different from the primary organ. This process requires genetic and epigenetic changes in cancer cells and the concomitant modification of the tumor stroma to facilitate invasion by metastatic cells. In this study, we analyzed differences in the epigenome of brain metastasis from the colon (n = 4) and lung (n = 14) cancer and we compared these signatures with those found in primary tumors. Results show that CRC tumors showed a high degree of genome-wide methylation compared to lung cancers. Further, brain metastasis from lung cancer deeply activates neural signatures able to modify the brain microenvironment favoring tumor cells adaptation. At the protein level, brain metastases from lung cancer show expression of the neural/glial marker Nestin. On the other hand, colon brain metastases show activation of metabolic signaling. These signatures are specific for metastatic tumors since primary cancers did not show such epigenetic derangements. In conclusion, our data shed light on the epi/molecular mechanisms that colon and lung cancers adopt to thrive in the brain environment.

## 1. Introduction

Brain metastases are the most common type of intracranial tumors in the adult population and occur in approximately 20% of all cancer patients [1]. They can cause a range of neurological symptoms, including headaches, neurological deficits, and cognitive impairment, which can significantly reduce a patient’s quality of life [2]. Moreover, the prognosis for cancer patients with brain metastases is generally poor and the incidence is increasing [3,4]. The underlying mechanisms through which cancer cells spread into the brain are still under investigation. Therefore, understanding the biological basis of the metastatic process to the brain is crucial for establishing prognostic/predictive algorithms and developing novel therapeutic approaches [5]. In fact, unraveling the molecular mechanisms that allow cancer cells to thrive in a new environment could enable us to develop specific targeted therapeutic approaches that could block or reverse such mechanisms, potentially resulting in metastasis regression [6]. There are complex components that contribute to brain invasion from the primary tumor, suggesting a common mechanism through which the tumor can metastasize. Particularly, epithelial-to-mesenchymal transition (EMT), aberrant TGF-β, Notch and β-catenin pathways activation and the deregulation of some microRNAs have a critical role in cerebral metastases [7]. EMT is crucial for cancer cell dissemination. During this process, cells lose their cell-cell contact and acquire a mesenchymal phenotype, driven by the expression of some mesenchymal markers, such as Nestin and Vimentin in lung cancer [8,9,10]. In addition, the ability of cancer cells to acquire additional mutations, and the interaction with the tumor microenvironment and brain niche may contribute to the site-specific dissemination of metastatic tumor cells [11,12,13]. In this regard, the epigenetics of brain metastases is a promising field of study. It has become increasingly clear that epigenetic mechanisms, such as histone modifications, non-coding RNA expression, and DNA methylation, are crucial for cancer cell survival in the metastatic environment. Alterations in DNA methylation, which result in alterations in DNA transcription and, therefore, protein expression, are exploited by metastatic cancer cells to adapt to the peculiar central nervous system microenvironment [14,15]. Altered DNA methylation and histone modifications have been recognized to be critical steps in cancer cell dissemination. Hypermethylation of several gene promoters has been associated with metastatic cell homing in the central nervous system, including *CXCR4* in breast cancer [16] and *PTEN* in melanoma [17]. Epigenetic therapies (“epidrugs”) are a promising approach for the treatment of advanced-stage malignancies, including brain metastases [18,19]. However, the drivers of these epi-events have not yet been fully identified.

Recent advances highlight the efficacy of oncolytic viruses (OVs) in combination with epigenetic modulators in cancer treatment, at both pre-clinical and clinical levels [20,21]. It has been demonstrated that OVs co-treatment with the histone deacetylase inhibitor (HDACi) Trichostatin A (TSA) reduced the number of metastases in lung cancer [22]. Similarly, the combination of OVs and Valproic Acid (VPA; HDACi) induced cell cycle arrest and apoptosis in cervical and pancreatic adenocarcinoma cell lines [23]. Thanks to the tumor cell selectivity and the ability to activate an anti-tumor immune response, OVs could represent a promising anticancer therapy, including disseminated metastasis

Here, we investigated the differentially methylated regions (DMRs) of brain metastases from colorectal and lung cancers to preliminary characterize the complex epigenetic interplay between metastatic cancer cells and resident brain cells.

## 2. Materials and Methods

### 2.1. Patients

For methylation analysis, patients with a diagnosis of non-small cell lung (LuC; n = 14) and colorectal (CRC; n = 4) cancer brain metastases (BM) were included in the study (detailed in Table 1). To exclude the presence of brain tissue in the sample, hematoxylin/eosin staining on frozen sections was performed.

For immunohistochemical analysis (IHC), 47 formalin-fixed and paraffin-embedded (FFPE) samples were included in a Tissue Micro-Array (TMA), including brain metastases from LuC (n = 41), one CRC, and five associated primary LuC. TMA sample information is detailed in Table 2. For each patient, one spot of non-neoplastic parenchyma and two tumor spots were included in the TMA as previously described [24]. Nestin IHC staining was performed with a monoclonal antibody (MAB1239, R&D, Minneapolis, MN, USA), and the Benchmark Ultra Roche Ventana immunostainer (Roche Group, Tucson, AZ, USA). Immunoreactivity was scored as the positive percentage of tumor cells in each core; the mean score between the two tumor cores was used for subsequent analyses.

### 2.2. DNA Purification and Bisulfite Conversion

Genomic DNA from frozen samples was extracted with the DNAeasy Blood and Tissue Kit (Qiagen), according to the manufacturer’s instruction and quantified using the Qubit dsDNA BR Assay kit ((Thermo Fisher Scientific, Waltham, MA, USA). DNA integrity was evaluated with the Agilent TapeStation 4200 (Agilent Technologies, Santa Clara, CA, USA). A DIN (DNA Integrity Number) > 6 was considered suitable for the analysis. Then, 500 ng of high-quality genomic DNA were bisulfite-converted using the EZ DNA Methylation-Gold Kit (Zymo Research, Irvine, CA, USA), according to the manufacturer’s procedure.

### 2.3. Infinium MethylationEPIC BeadChip Array

The sample DNA methylation profile was analyzed using the Infinium MethylationEPIC Kit (Illumina, San Diego, CA, USA), following the manufacturer’s protocol. Briefly, 4 μL of bisulfite-converted DNA was amplified, fragmented and precipitated. The DNA was subsequently hybridized to probes on the BeadChips and scanned using the NextSeq 550 System (Illumina). A raw .idat file was generated and imported into Genome Studio software v2.0.5 to check technical parameters quality control, including bisulfite conversion efficiency, dye specificity, hybridization and staining.

### 2.4. Data Analysis

DNA methylation of metastatic samples (n = 18) was assessed using the Infinium HumanMethylation450 BeadChip (Illumina). Moreover, Infinium data from 350 primary colon cancer and 507 primary lung cancer samples were downloaded, respectively, from TCGA-COAD (https://portal.gdc.cancer.gov/projects/TCGA-COAD, accessed on 23 January 2023) and from TCGA-LUAD (https://portal.gdc.cancer.gov/projects/TCGA-LUAD, accessed on 23 January 2023) projects.

Analysis of methylation data was performed by mirroring the workflow published by Maksimovic and collaborators [25]. All public and in-house-generated methylation data were merged for a total of 875 samples and 452,832 probes. Accordingly, a *p*-value (cut-off 0.05) was calculated for every CpG in every sample using the detectionP function by comparing the total signal of each probe to the background, which was estimated from the negative control probes. Poor-quality samples with a detection *p*-value > 0.05 were excluded from the analysis. Normalization was performed using the Functional normalization by FunNorm R function of minfi package removing unwanted variation by regressing out variability explained by the control probes present on the array. Poor-performing probes, probes from the X and Y chromosomes, probes that are known to have common SNPs at a CpG site and cross-reactive probes were filtered out.

In the end, our data matrix consisted of 334,097 probes and 875 samples and M-values (generated by Mvalue function) was used for subsequent statistical analyses. Differential methylation analysis of regions and annotation were performed to assess whether several proximal CpGs were concordantly differentially methylated (differentially methylated region, DMR). Gene Ontology (GO) analysis using the R package ClusterProfiler was then performed with significant DMR to gain insights into potentially affected biological processes. The web-based Metascape tool (https://metascape.org/, accessed on 24 February 2023) was used to relate GO terms in networks, where terms with a similarity > 0.3 are connected by edges. Unsupervised PCA analysis was performed as previously described [26]. Venn diagrams were generated using the Bioinformatics and Evolutionary Genomics online tool [27], while other charts were drawn using GraphPad Prism software V7.

## 3. Results

### 3.1. CpG Methylation Pattern in Brain Metastasis from CRC or Lung Cancers

We started this study by analyzing methylation patterns in brain metastases compared with primary tumors. CRC and lung cancers displayed distinct epigenetic signatures (Appendix A). Indeed, brain metastasis from CRC preferentially hypermethylated gene promoters compared with primary tumors whilst brain metastasis from lung cancer hypomethylated gene promoters compared with primary LuC. We then performed a Gene Ontology analysis with only DMRs that show an adj. *p* value < 0.0001 (Appendix A). Hypermethylated DMRs in BM-CRC affected biological processes involved in DNA replication, gene translation and cell cycle (Figure 1A). On the contrary, brain metastasis from lung cancer showed activation (i.e., hypomethylation) of signaling involved in embryonic development and neurodevelopment (Figure 1B). BM-CRC did not hypomethylate the region below our threshold of adj. *p* < 0.0001; we could detect few DMRs with an adj. *p* value < 0.05 (n = 13; Appendix A), whereas BM-LuC did not significantly hypermethylate any DMR. We then compared GO annotation from significant hypomethylated DMRs in BM-CRC and BM-LuC. No element was shared between the two metastatic diseases (Figure 1C). Further, unsupervised principal component analysis (PCA; Figure 1D) of epigenetic traits showed that brain metastasis from CRC and LuC have distinct patterns of methylation. These results suggest that the two cancers adopt different epigenetic changes to metastasize to the brain.

### 3.2. Differentially Methylated Regions Specific for CRC and LuC brain Metastasis

Next, we compared DMRs in CRC and LuC primary tumors (Appendix A) and in the metastatic setting (Appendix A), and we performed a GO analysis (Appendix A) to see whether we could gain insights into proprietary epigenetic signatures of primary cancers and of brain metastases. Functional analyses of BM-type-specific hypomethylated regions revealed relevant genes and pathways specific to CRC and LuC brain metastasis. LuC tumors showed activation of 295 signaling (GO terms) in primary samples, 458 in BM, and 222 (75%) GO terms were concordantly upregulated in primary and BM-LuC (Figure 2A).

On the contrary, functional analysis of hypomethylated DMRs in CRC compared with LuC samples showed activation of only 107 signaling in primary CRC, 49 in BM-CRC, and 14 (13%) GO terms were concordantly activated in CRC and BM-CRC (Figure 2B). This result suggests that CRC opposite to LuC tumors preferentially hypermethylate their epigenome and these traits are conserved in the primary and metastatic settings. Further, in CRC the quota of hypermethylated regions increases in brain metastasis compared to the primary tumor opposite to what is observed in LuC (Appendix A).

Annotation of hypomethylated DMRs in lung neoplasms showed that the development/neurodevelopment signaling was the most activated process (Appendix A), and among activated signaling in CRC neoplasms there were cell transport and cell adhesion (Appendix A). These signalings were retained in primary and metastatic cancers.

To provide preliminary evidence of tumor-origin specific epigenetic signature of brain metastases, we then “subtracted” DMRs found in primary tumors from those identified in the metastatic setting (Appendix A) and we annotated the resulting DMRs as BM-specific (Appendix A). Using this approach, we found that BM-LuC is characterized by the activation of neurological signaling (Figure 2C,E and Appendix A), while BM-CRC shows the potential activation of metabolic processes (Figure 2D,F and Appendix A).

### 3.3. Brain Metastases from Lung Cancer Acquire Neuronal Markers

Although preliminary and partially in silico, the above data pointed to the acquisition of neurological/neurodevelopmental traits by brain metastases from lung cancer. To test this hypothesis at the phenotypic level we stained an independent archival series of brain metastasis [24] with the neuronal marker Nestin and we scored the percentage of positive cancer cells.

Among BM-LuC, Nestin was positive in 31/40 cases (78%; one case did not show sufficient cancer cells for evaluation), with a mean expression of 27% (range 0–100%; Figure 3A,B). Primary LuC (n = 5) showed a modest Nestin expression (0.5–1%) in 2 out of 5 cases (Figure 3A,B). The only BM from CRC retrieved from our archive showed Nestin expression in 12% of neoplastic cells (Figure 3A,B).

## 4. Discussion

Tumor cell plasticity and the rewiring of cell signals are controlled by genetic and epigenetic derangements of regions involved in cancer cell survival, growth and metastasis. Little is known about the epigenetic signatures governing the dissemination of a tumor from the primary site to the brain. Our study, although preliminary, highlights epigenetic differences between CRC- and LuC-derived brain metastasis. Despite both cancers can colonize the brain as a distant site, the epigenetic changes underpinning this process are distinct. Indeed, metastatic CRC preferentially hypermethylated its epigenome whilst LuC hypomethylated it. In terms of potentially activated signatures (i.e., hypomethylated regions), brain metastasis from CRC hypomethylates genes involved in cell metabolism. On the other hand, brain metastasis from LuC activates neural signaling. This preliminary observation was supported by the immunophenotypic evaluation of Nestin expression in archival tissues from primary and metastatic LuC samples.

Hypermethylation in CRC, also known as the CpG island methylator phenotype (CIMP), has already been described in primary tumors [28] and the CRC epigenome has been demonstrated to facilitate the accumulation of genomic mutations in cancer key genes [29]. The CIMP-high CRC phenotype has been associated with poor prognosis and cancer aggressiveness [30]. Our data confirm this notion providing that brain metastasis from CRC maintains the hypermethylated status observed in primary tumors and shows modest hypomethylation at metastasis-specific regions, i.e., promoters of genes involved in metabolic pathways. Further, we could confirm that the epigenome can accurately distinguish brain metastasis according to the cell of origin [31,32] since BM-CRC and BM-LuC were clearly separated in the unsupervised PCA analysis. The epigenetic analysis could, therefore, potentially support the diagnosis and therapeutic approach of BM of unknown origin.

Opposite to CRC, LuC showed a tendency to hypomethylate its epigenome and activate neural-related signaling. Importantly, our analysis shows that embryonic and neurodevelopmental pathways are already activated in primary LuC and retained in the metastatic setting. Further, BM-LuC specifically hypomethylate (i.e., overexpress) another small, yet potentially important, set of genes involved in neural signaling. This result may explain the high frequency of BM in LuC patients, and the success of LuC cells to thrive and adapt in the brain milieu. Our data are in line with a recent observation [33] and expand our understanding of the epigenetic landscape of BM-LuC showing the acquisition by BM of the neural marker Nestin. Different studies showed how an altered DNA damage response, the homologous recombination and mismatch repair deficiency signatures govern the brain metastatic process in CRCs [34]. Moreover, CRC metastatic cells deriving from the primary tumor are characterized by hypermethylation on the promoter of E-cadherin, promoting cell-cell contact loss and EMT progression [35]. Evidence of epigenetic reprogramming of metastatic lung cancer to the brain has already been highlighted [33]. In this study, the authors found similar variations in the epigenetic pattern in matched primitive and metastatic cancers compared to non-neoplastic lungs. They described that the hypermethylation of DNA methylation valley (DMV) regions was a characteristic of metastatic lesions. In line with our findings, they show that most of the differentially methylated DMVs harbored development genes and they conclude that the fine-tuning by epigenetic mechanisms of such genes may confer a selective advantage to metastatic lung cells to colonize the brain. In keeping with this observation we found expression of Nestin by BM-LuC and not by *p*-LuC tumors [33].

Weaknesses of this study include the small sample size, the preliminary nature of the analysis, and the lack of genetic characterization of the BM samples, including the assessment of chromosomal or microsatellite instability. Particularly, we could retrieve only a few BM-CRCs compared to BM-LuC. The rarity of brain metastasis from this histological entity, and the late-stage nature of this event may account for the limited size of the cohort. Despite the above limitations, our study offers an overview of the epigenetic differences of brain metastasis from CRC and LuC cancers. Further studies with larger cohorts are needed to confirm our preliminary data.

## 5. Conclusions

We could provide specific clues for CRC and LuC-derived BM, activated signaling and a preliminary validation at the protein level of the observed acquisition by BM-LuC of neural immunophenotypes. Epigenetic changes increase cancer cell fitness and plasticity, thereby fostering cell adaptation to foreign environments and stimuli. On the other side, epigenetic changes can be reversed. Therefore, a deeper knowledge of the epigenetics of brain metastases will not only help the diagnosis of the tumor of origin but will increase our understanding of how cancer progresses and spreads, providing new clues for BM treatment which is currently lacking in the clinical setting.

## Figures and Tables

**Figure 1 cancers-15-02145-f001:**
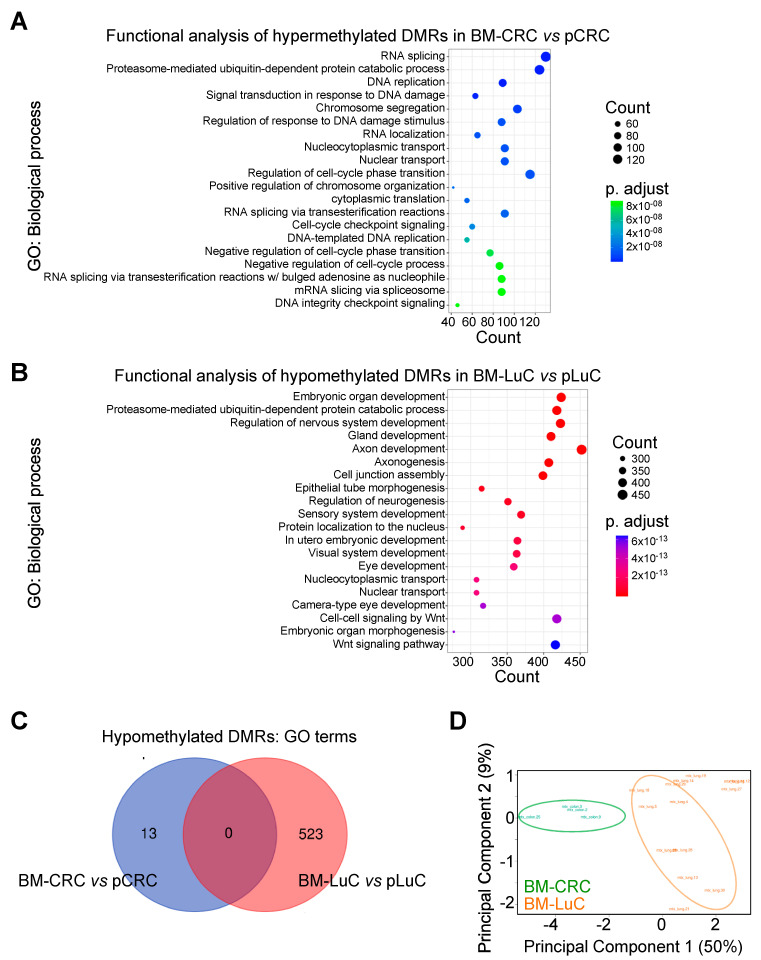
Epigenetic signature of brain metastasis from CRC and LuC tumors. (**A**,**B**) Differentially methylated regions (DMRs) between metastatic and primary tumors were analyzed and annotated using Gene Ontology (GO)-biological process tool. Graphs show the top 20 identified signatures. (**C**) GO terms relative to hypomethylated DMRs (i.e., activated signatures) in brain metastasis compared with primary tumors were compared using a VENN diagram. (**D**) Principal Component Analysis (PCA) of the methylome of brain metastases from CRC and LuC tumors. The variance explained by the first two components is shown in brackets. pCRC, primary CRC; BM-CRC, brain metastasis from CRC; pLuC, primary lung cancer; BM-LuC, brain metastasis from lung cancer.

**Figure 2 cancers-15-02145-f002:**
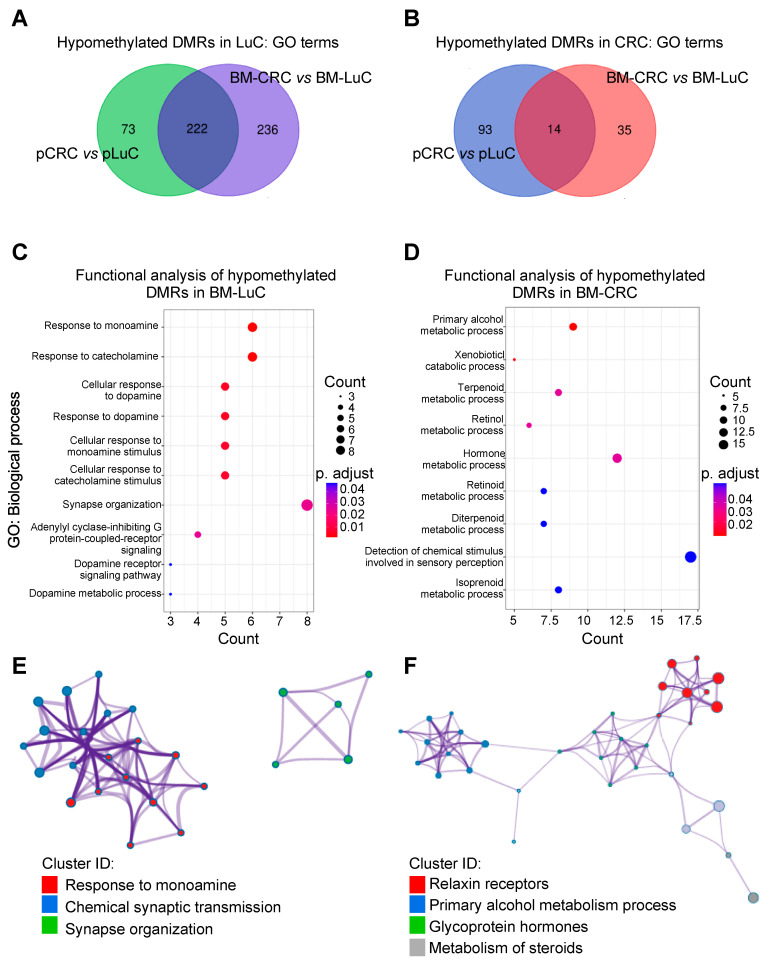
Identification of tumor-type specific epigenetic signatures of brain metastasis. (**A**,**B**) DMRs associated with the tumor-type were identified in primary and metastatic cancers and functionally annotated using GO. Then, VENN diagrams were used to identify shared activated signaling (i.e., hypomethylated DMRs) in LuC (**A**) and CRC (**B**) neoplasms. (**C**,**D**) DMRs selectively hypomethylated in BM-LuC (**C**) or BM-CRC (**D**) were annotated using the GO-Biological process tool. (**E**,**F**) Networks of enriched terms in BM-LuC (**E**) or BM-CRC (**F**) were colored by cluster ID using the Metascape tool. Nodes that share the same cluster ID are connected. pCRC, primary CRC; BM-CRC, brain metastasis from CRC; pLuC, primary lung cancer; BM-LuC, brain metastasis from lung cancer.

**Figure 3 cancers-15-02145-f003:**
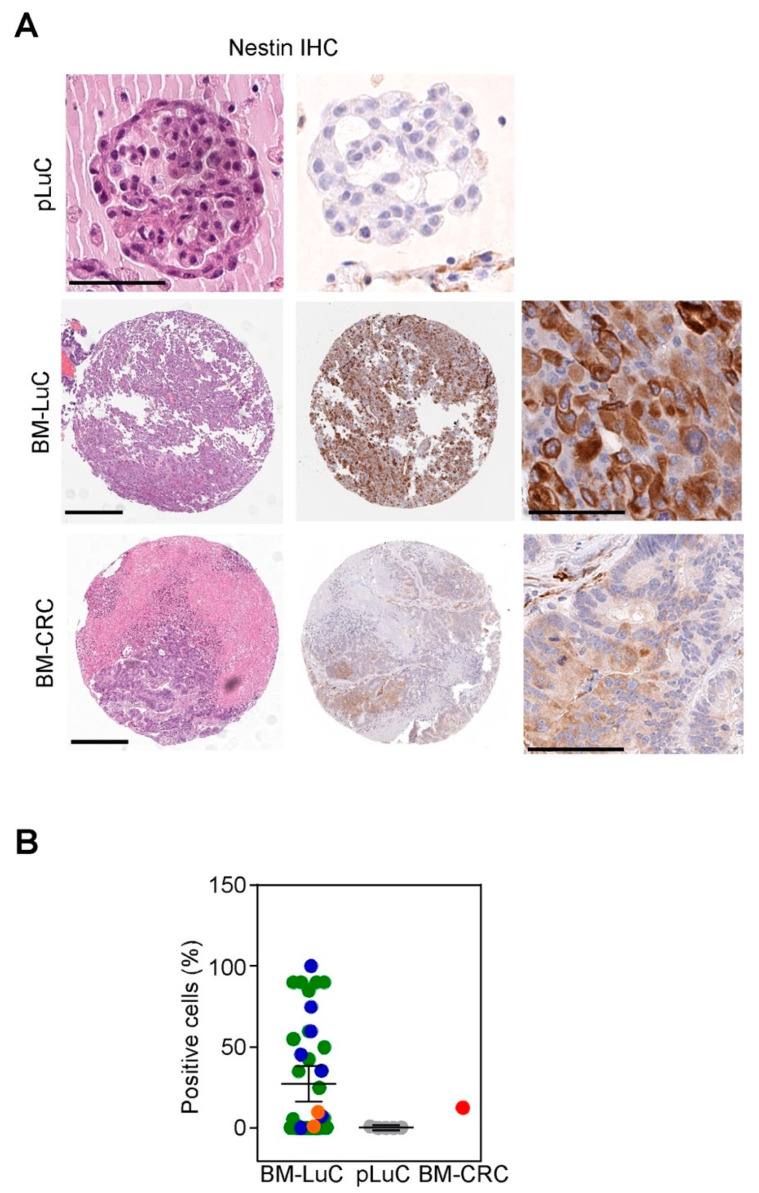
Immunophenotypic characterization of BM-LuC for the neural marker Nestin. (**A**,**B**) Nestin immunohistochemistry (IHC) was performed in an independent series of archival BM from lung (n = 41; blue dots SCC histology, green dots ADCA histology, orange dots NSCLC-NOS histology) or CRC (n = 1) cancers and in five primary LuC. Representative images (**A**) are shown for a pLuC, BM-LuC, and BM-CRC (scale bar, 100 μm). Nestin expression was quantified in cancer cells and expressed as a percentage of positive cancer cells (**B**).

**Table 1 cancers-15-02145-t001:** Histological characteristics of the brain metastases included in the methylome analysis.

				Gene Mutation						Immunephenotype							
ID	Gender	Age at Dx	Histology	ALK-R	BRAF	EGFR	ERBB2	KRAS	ROS1-R	CK7	CKAE1/3	NAPSIN-A	p40	p63	TTF1	CDX2	CK20
	Lung																
BM-LuC1	M	78	NSCLC, NOS			WT				+	+	-		-	-		-
BM-LuC2	M	73	ADCA			WT						+	-		+		
BM-LuC3	M	60	SCC							-			+				-
BM-LuC4	M	57	ADCA	WT		WT			WT	+		+	-		+		-
BM-LuC5	F	70	ADCA	WT		WT			Mut	+		+			+		-
BM-LuC6	M	55	ADCA	WT	WT	WT		WT		+		-			+	-	-
BM-LuC7	M	54	ADCA	WT		WT			WT	+		+			+		-
BM-LuC8	F	65	ADCA							+		+	-		+		-
BM-LuC9	M	50	NSCLC, NOS	WT		WT											
BM-LuC10	M	43	NSCLC, NOS	WT	WT	WT	Mut	WT	WT	+		-	-	-	-		-
BM-LuC11	F	59	SCC										+				
BM-LuC12	M	75	SCC									-		+			
BM-LuC13	F	62	ADCA	Mut		WT				+		-			-	-	-
BM-LuC14	M	69	SCC	WT		WT			WT	-	+		+		-		-
	Colon																
BM-CRC1	M	62	ADCA													+	+
BM-CRC2	M	65	ADCA							-						+	+
BM-CRC3	F	71	ADCA		WT			Mut		-						+	+
BM-CRC4	F	80	ADCA													+	+

BM: brain metastases; LuC: Lung cancer; CRC; colorectal cancer; M: male; F: female; Dx, diagnosis; ADCA: adenocarcinoma; SCC: squamous cell carcinoma; NSCLC, NOS: non-small cell lung carcinoma, not otherwise specified; WT, wild-type; Mut, mutated; empty cell, assay not performed; -, absence of staining; +, presence of staining.

**Table 2 cancers-15-02145-t002:** Characteristics of the tumor samples included in the TMA analysis.

Primary	Histology	N
Metastases		
Lung	ADCA	32
	SCC	2
	NSCLC-NOS	7
Colon	ADCA	1
Primary cancer		
Lung	ADCA	5

TMA, tissue microarray; ADCA, adenocarcinoma; SCC, squamous cell carcinoma; NSCLC-NOS, non-small cell lung carcinoma, not otherwise specified.

## Data Availability

Data supporting reported results can be found within the main manuscript or in the Appendix A. Methylome raw data will be deposited to a publicly database upon manuscript acceptance.

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
