# Peer review of "Epigenetic Rewiring of Metastatic Cancer to the Brain: Focus on Lung and Colon Cancers"

_cancers, 2023, doi:10.3390/cancers15072145_

Round 1

Reviewer 1 Report

In the manuscript “Epigenetic rewiring of metastatic cancer to the brain: focus on 2 lung and colon cancers” the authors study epigenetic changes in tumor cells focusing on differences in the epigenome of brain metastases from colon and lung cancer.

The work is well structured, and all sections are described correctly. I would just suggest to the authors to refer to the paper written by Chianese et al (Oncolytic viruses in combination therapeutic approaches with epigenetic modulators: past, present and future perspectives; PMID:34199429) which shows how many tumors are treated with epigenetic modulators in combination with oncolytic viruses (in lines 70-72).

Author Response

Reviewer#1 comments:

The work is well structured, and all sections are described correctly. I would just suggest to the authors to refer to the paper written by Chianese et al (Oncolytic viruses in combination therapeutic approaches with epigenetic modulators: past, present and future perspectives; PMID:34199429) which shows how many tumors are treated with epigenetic modulators in combination with oncolytic viruses (in lines 70-72).

We thank the Reviewer for his/her comment. Accordingly, we inserted the suggested reference (#20) and we commented the topic in the Introduction (adding refs#21,22).

Reviewer 2 Report

The Research Article titled "Epigenetic rewiring of metastatic cancer to the brain: focus on lung and colon cancers", authored

by Annamaria Morotti and colleagues presents differences in the epigenome of brain metastases from colon and lung cancer and compares these signatures with those found in primary tumors. 

Overall, the study is well designed and the results are clearly presented. 

The study's conclusions are supported by the data presented, and the authors' interpretations of the results are logical and well-supported.

The use of primary tumors as a reference provides a useful context for interpreting the results of the study. 

My comments are following:

1) The sample size for colon cancer metastases is relatively small. The authors acknowledge this limitation but should highlight the need for future studies with larger sample sizes. A paragraph with study limitations including sample size should be added in the discussion.

2) The discussion section lacks a thorough presentation of the important findings of the study compared to the literature. I would suggest an expansion of a couple of paragraphs.

3) I believe some demographics data, if available, would be informative and should be added in Table 1.

Author Response

Reviewer#2 comments:

1) The sample size for colon cancer metastases is relatively small. The authors acknowledge this limitation but should highlight the need for future studies with larger sample sizes. A paragraph with study limitations including sample size should be added in the discussion.

As suggested, we added a paragraph highlighting study’s limitation at the end of the discussion.

2) The discussion section lacks a thorough presentation of the important findings of the study compared to the literature. I would suggest an expansion of a couple of paragraphs.

As suggested, we discussed our data compared to what is already known (Discussion section).

3) I believe some demographics data, if available, would be informative and should be added in Table 1. We thank the Reviewer for this suggestion. Accordingly, we revised Table 1 adding some demographic information together with the molecular and immunophenotypic characterization of the brain metastases samples.

Reviewer 3 Report

This study compared the epigenetic differences between brain metastases derived from colorectal cancer (CRC) and lung cancer (LuC). The results showed that the epigenetic changes underlying this process are distinct, with CRC preferentially hypermethylating its epigenome while LuC hypomethylated it. The study highlights the potential of epigenetic analysis in supporting the diagnosis and therapeutic approach of brain metastases of unknown origin. The study expands our understanding of the epigenetic landscape of brain metastases, which can provide new clues for the development of effective treatments for this devastating complication of cancer. However, the study has some limitations which needs to be addressed:

1. The study did not include a genetic characterization of the brain metastasis samples, such as the assessment of chromosomal or microsatellite instability. This information could provide additional insights into the underlying mechanisms of brain metastasis.

2.While the study provides evidence of epigenetic changes and gene expression patterns associated with brain metastasis, it did not perform functional experiments to validate these findings. Additional experiments such as knockdown or overexpression of specific genes could help to confirm their role in brain metastasis.

3. The study only compared brain metastasis from CRC and LuC, without including a control group. Adding a group of patients without brain metastasis from these cancer types could help to identify epigenetic changes specific to brain metastasis rather than the primary tumor.

4. The sample size for some analyses, especially for CRC brain metastases, is relatively small, which limits the generalizability of the findings.

5. The study did not account for potential confounding factors such as age, sex, and lifestyle factors, which may affect methylation patterns and introduce bias in the results.

6. The study only included samples with an adj. p value < 0.0001, which may have excluded important DMRs that could have contributed to the differences in methylation patterns between the two types of brain metastases.

Although, most of these limitations are mentioned by the authors but the study remains incomplete in the absence of these.

Author Response

Reviewer#3 comments:

  1. The study did not include a genetic characterization of the brain metastasis samples, such as the assessment of chromosomal or microsatellite instability. This information could provide additional insights into the underlying mechanisms of brain metastasis.
  2. While the study provides evidence of epigenetic changes and gene expression patterns associated with brain metastasis, it did not perform functional experiments to validate these findings. Additional experiments such as knockdown or overexpression of specific genes could help to confirm their role in brain metastasis.

We agree with the Reviewer that a more comprehensive genetic analysis of brain metastasis together with functional experiment (other than the immunohistochemical evaluation of Nestin) would extend the preliminary knowledge provided within this manuscript. Nevertheless, these experiments, in our opinion, could be the focus of a next paper aimed at validating our preliminary and exploratory observation. We hope that Editors and the Reviewer would understand that the suggested experiments would require a large amount of time, specific reagents (cell lines representative of CRC or LuC brain mets), and money to be performed and therefore cannot be included in the present work.

  1. The study only compared brain metastasis from CRC and LuC, without including a control group. Adding a group of patients without brain metastasis from these cancer types could help to identify epigenetic changes specific to brain metastasis rather than the primary tumor.

We completely agree with the Reviewer, indeed we included two series of CRC and LuC primary cancers and we used as controls to identify lineage-specific or metastasis-specific epigenetic signatures in the two tumor settings. Results from primary cancers are presented in Figures 1, 2A,B and Supplementary Tables 1-3, 7 and 8.

  1. The sample size for some analyses, especially for CRC brain metastases, is relatively small, which limits the generalizability of the findings.

We concur with the Reviewer but fresh sample from brain metastases are a rarity and we included in the study all available samples. We acknowledge this limitation in the discussion section.

  1. The study did not account for potential confounding factors such as age, sex, and lifestyle factors, which may affect methylation patterns and introduce bias in the results.

We compared the prevalence of the aforementioned factors in the primary (pLuC and pCRC) and metastatic CRC and LuC (BM-CRC and BM-LuC) series and calculated the pvalues resulting from the chi-squared test. This analysis (provided below) did not show differences in the patients’ series; therefore, p values were not adjusted.

pLuC_ TCGA

LuC_ BM

p val (chi-squared test)

n

507

14

M/F ratio

0.9

2.5

p=0.207

Age at Dx

65.3+/-10.3

62.1+/-10.1

pCRC_TCGA

CRC_BM

n

350

4

M/F ratio

1.1

1.0

Age at Dx

66.1+/-13.2

69.5+/-7.9

  1. The study only included samples with an adj. p value < 0.0001, which may have excluded important DMRs that could have contributed to the differences in methylation patterns between the two types of brain metastases.

We concur with the Reviewer and accordingly we included all resulting DMRs (also with a non-significant p value) in the Supplementary Tables 9, 10, 15 and 15.

Round 2

Reviewer 3 Report

I am satisfied with authors response.